# Dietary Fat Intake among Chinese Adults and Their Relationships with Blood Lipids: Findings from China Nutrition and Health Surveillance and Comparison with the PURE Study

**DOI:** 10.3390/nu14245262

**Published:** 2022-12-09

**Authors:** Rongping Zhao, Liyun Zhao, Fan Yang, Lahong Ju, Shujuan Li, Xue Cheng, Xiaoli Xu, Qiya Guo, Shuya Cai, Hongyun Fang, Dongmei Yu, Gangqiang Ding

**Affiliations:** 1Chinese Center for Disease Control and Prevention, National Institute for Nutrition and Health, Beijing 100050, China; 2National Cancer Center/National Clinical Research Center for Cancer/Cancer Hospital, Chinese Academy of Medical Sciences and Peking Union Medical College, Beijing 100050, China

**Keywords:** dietary fats, dyslipidemia, Asians

## Abstract

Dietary fat intake in the Chinese population has increased. The PURE (prospective urban rural epidemiology) study concluded the potential advantage of total fat and saturated fats (SFA) over carbohydrates on blood lipids with the inaccurate assessment of dietary fats. We investigated the dietary fat profile among 48,315 participants (aged 30–70 years, national representative) from the China Nutrition and Health Surveillance (2015–2017), determined the relationship of various fats with blood lipid biomarkers in the selected 39,115 participants, and compared the results with the PURE study. Dietary fat intake was calculated by using 3 days of 24 h recalls and food inventory. Serum lipid biomarkers were assessed at morning fasting state by health professionals. Plant fats (21.5% of total energy (TE)) dominated in total fat intake (69.5 g/day, 35.6% TE), with monounsaturated fats (MUFA) in the largest (13.8% TE) portion and plant oils as the major source (43.7%). Two thirds of the population consumed more than 30% TE from dietary fats and nearly half more than 35%, while 26.4% of them exceeded 10% TE from SFA. Total fat was positively associated with total cholesterol (TC), low-density lipoprotein cholesterol (LDL-c), but also high-density lipoprotein cholesterol (HDL-c), and negatively with triglyceride (TG)-to-HDL-c ratio (TGHDL) (all *p*_-trend_ < 0.05). Replacing total fat with carbohydrate yielded adverse changes in most biomarkers (higher LDL-c, TG, and TGHDL, lower HDL-c, all *p* < 0.05) when total fat was low (<34.9% TE). Regardless of fat intake level, replacing plant fat or polyunsaturated fats (PUFA) with carbohydrate yielded the most adverse changes (higher TC, LDL-c, TG, TC-to-HDL-c ratio (TCHDL), and TGHDL, but lower HDL-c, all *p* < 0.01), while replacing animal fat/SFA with plant fat/PUFA yielded the most favorable changes (lower all biomarkers, all *p* < 0.01). The results suggested a less harmful effect of carbohydrate on blood lipids when total fat was high, and more beneficial effect of unsaturated fats, than the PURE. In conclusion, dietary fat intake in Chinese adults had reached quite a high level, but with a different profile from Western populations. Replacement of SFA (animal fat) with PUFA (plant fat) could most effectively improve blood lipids, while replacement with carbohydrate could slightly benefit only when total fat was high. The present results may be more applicable to the Chinese population than the PURE study.

## 1. Introduction

The global epicenter of non-optimal cholesterol is shifting to low- and middle-income countries, with China ranked as the top countries with the largest magnitudes of increase in mean non-high-density lipoprotein (HDL) cholesterol [1,2]. Along with the surge of dyslipidemia was remarkably increased prevalence and a heavy burden of cardiovascular diseases in China [3]. As a consequence of dietary and behavioral changes, dyslipidemia, usually diagnosed by multiple lipid biomarkers which have mutually independent effects on cardiovascular diseases and mortality, has long been suggested to be primarily managed by modifying dietary intakes, which included limiting total fat and saturated fatty acids (SFA) intake to less than 35% of total energy (TE) and 10% TE, respectively [4,5,6]. Without the assessment of plant oils intake, which is the major source of dietary fats in Chinese adults, the PURE (prospective urban rural epidemiology) study suggested the potential beneficial effect of total fat and the advantage of SFA over carbohydrates on blood lipids [7,8]. Additionally, more and more evidence, from Western countries predominantly, suggests the quality of dietary fats, especially food sources of SFA and monounsaturated fatty acids (MUFA), might be more important than quantity in their relations to blood lipids and cardiovascular diseases, with most approval to plant sources and disapproval to animal sources of fats [9,10,11,12,13,14,15].

The traditional high-carbohydrate diet in the Chinese population is rapidly Westernized with upward fat intake, but the dietary fat intake profile, particularly of various types and food sources, was not fully illustrated with national representative data [8,16,17]. Surprisingly, few studies systematically investigated the effect of dietary fats, particularly of various types and sources, on multiple lipid biomarkers in the Chinese population, although one study conducted in north China had linked dietary patterns with a single lipid marker of HDL cholesterol (HDL-c), and another study with national samples linked increased pork intake with hypercholesterolemia [18,19]. To fill the gap, we aimed to study the profile of dietary fat intake of the Chinese population and comprehensively investigate the relationships of dietary fats with lipid biomarkers across various types and sources of dietary fats using data from China Nutrition and Health Surveillance (CNHS) (2015–2017). Our second aim was to examine whether these relationships vary by intake level, and to assess the replacement effect of various fats with carbohydrate/other fats on lipid biomarkers and compare them with the results of the PURE study.

## 2. Materials and Methods

### 2.1. Study Population

Data used for analysis were from the CNHS (2015–2017), a national surveillance periodically conducted by the Chinese Center for Disease Control and Prevention (CDC). The present study used data of adult surveillance in 2015. Details about designs, sampling methods, and data collection of the surveillance had been described elsewhere [20,21]. Briefly, the CNHS (2015–2017) covered 298 survey sites across 31 provincial administrative units in mainland China. A stratified, multistage, probability-based random sampling scheme was used to select eligible participants, who were aged 18 years and older, living in the sample area for more than 6 months during last 12 months. The targeted population of this cross-sectional study was adults aged between 30 and 70 years (48,693 cases in total). The whole sample (48,315 cases) for analysis of dietary fat profiles among the targeted population excluded 378 participants with abnormal energy intake (<500 kcal or >4000 kcal/day), while the selected sample (39,115 cases) for analysis of the relationships between dietary fats and blood lipids further excluded participants with diagnosed cardiovascular diseases (CVD), cancer (5590 cases), dyslipidemia (2456 cases), or diabetes (1154 cases). 

### 2.2. Assessment of Dietary Fats

Diet was assessed using 3 days (2 weekdays and 1 weekend) of 24 h dietary recalls in addition to weighing household cooking oils and condiments. For each dietary recall day, trained investigators went to the participant’s home and helped to recall and record food items and intakes (eaten in home and outside) during the last 24 h. Investigators also weighed the household cooking oil and condiments, and recorded the number of diners (including guests) at each meal in the home as well as their sex, age, and physical activity level at the beginning and end of each 24 h survey. This information was used to allocate the 3-day consumption proportion of cooking oils and condiments for each participant in the household, and calculate their actual number of meals and consumption during the 3 days. 

Each food item (including cooking oils and condiments) was coded according to the constantly updating Chinese Food Composition Database (https://www.phsciencedata.cn/Share/ky_sjml.jsp?id=577e0301-ab65-432a-9bb7-a8342302e589, accessed on 7 September 2022), classified as animal or plant source according to the major food composition, and further classified into 12 food groups, including pork, other meats, processed meats, poultry, fish, dairy and eggs, nuts and soybeans, oils of high MUFA, oils of high polyunsaturated fatty acids (PUFA), MUFA–PUFA balanced oils, oils of high saturated fatty acids (SFA), and other foods (for detailed definitions see Appendix A) [22,23]. Fat intakes of various types and sources were then calculated using the database.

### 2.3. Assessment of Blood Lipids

A fasting blood sample of the participants was collected by health professionals from the local CDC. Serum was separated from the blood and placed in the car refrigerator or box with ice cubes, delivered to the laboratory of the local CDC, and stored in the refrigerator at −20 °C within 2 h. Then, within one week after sample collection, samples were frozen at −80 °C and shipped by air with dry ice to the central laboratory in Beijing, China. Fasting blood samples were analyzed for total cholesterol (TC), low-density lipoprotein (LDL) cholesterol (LDL-c), HDL-c, and triglycerides (TG). The ratio of TC to HDL-c (TCHDL) and the ratio of TG to HDL-c (TGHDL) were also calculated. 

### 2.4. Covariates Assessment

Age, sex, ethnic group (Han, Zhuang, Manchu, Hui, Miao, Uygur, Yi, Tujia, Mongolia, Korean, Tibetan, and other ethnic), education level (illiteracy, primary school, middle school, high school, and college and above), physical activity, drinking behavior (never, moderate, or excessive (15–24.9 or ≥25 g/day)), smoking behavior (never, ever, or current (<10, 10–19, or ≥20 cigarettes/day)), and health information (family history of cardiovascular diseases and diabetes) were collected with individual questionnaires. Physical activity was measured with the global physical activity questionnaire (metabolic equivalent of task (MET)-h/week), and drinking behavior was collected by the food frequency questionnaire.

### 2.5. Statistical Analysis

Means (standard deviations (SDs)) or median (interquartile range (IQR)) were calculated to summarize continuous variables. Weights were calculated for estimation of population weighted dietary indicators. Methods to calculate weights were reported by a previous study [24]. The socioeconomic structure of the 2015 Chinese population estimated by the State Statistics Bureau (https://data.stats.gov.cn, accessed on 4 March 2022) was the basis for the post-stratification weights. General liner model was performed to test the differences and linear trends of dietary indicators among participants with different characteristics, and to estimate the relationship of various dietary fats with lipid biomarkers. The strength of associations between the various dietary fats and blood lipids indicators were compared in common units by standardized coefficients that represented the number of SDs a lipid marker changed per 1 SD increase in dietary fats. Participants with diagnosed hypertension were further excluded in sensitivity analyses. Low or high intake of dietary fat was categorized by the median of total fat intake, and stratified analysis was performed to assess the effect of fat intake level on the slope of association between various fat and lipid biomarkers. Nutrient residual model was used to estimate the effect of isocaloric replacement of various fats with carbohydrate or other fats [25]. In this procedure, the fat intakes of the individuals were regressed on their total energy intakes. The residuals, representing the differences of individuals’ actual intake independent of energy intake, plus the term of TE were both included in the models. Data analyses were performed with SAS version 9.4 (SAS Institute, Inc., Cary, NC, USA).

## 3. Results

### 3.1. Dietary Fats Intake Profile

The whole sample (mean (SD) age, 52.3 (10.2) years) had 26,696 women (55.2%), 28,394 (58.7%) rural residents, and 28,258 (58.5%) southern residents (Table 1). All dietary indicators with this sample were age-sex-standardized. Mean (standard error (SE)) of dietary energy, total fat, animal fat, SFA, MUFA, PUFA, and percentage energy from total fat, animal fat, SFA, MUFA, and PUFA were 1742.8 (13.2) kcal, 69.5 (0.7) g, 27.8 (0.7) g, 16.1 (0.2) g, 27.0 (0.4) g, 18.3 (0.2) g, 35.6% (0.3%), 14.1% (0.3%), 8.2% (0.1%), 13.8% (0.2%), and 9.5% (0.1%), respectively. About two thirds (66.8% (0.9%)) of the study population had more than 30% energy from dietary fats with nearly half (49.5% (1.1%)) more than 35% and only 24.2% (0.7%) within the recommended range of 20–30%; however, only 26.4% (1.1%) of them had more than 10% energy from SFA, and 16.4% (0.9%) of them had below 5%.

Percentage energy from animal fat and SFA decreased as age increased, yet percentage energy from total fat, animal fat, SFA, and MUFA increased as smoking cigarettes increased (*p*_-trend_ < 0.05). Higher percentage energy from animal fat and SFA presented in participants of men, urban, southern regions, and higher education level, and higher percentage energy from MUFA presented in participants of southern regions and higher education level, while higher percentage energy from PUFA presented in participants of women, northern regions, and lower education level (*p* or *p*_-trend_ < 0.05). Additionally, higher percentage energy from PUFA yet lower percentage energy from animal fat, SFA, and MUFA presented in participants with a family history and hypertension (*p*_-trend_ < 0.05).

Plant oils were the top source of total fats (43.7% (0.8%)), followed by pork (18.9% (0.5%)) which is the top source (50.8% (0.8%)) of animal fat (table not shown). Nearly half of SFA came from animal foods with pork being in the largest portion (30.4% (0.5%), Table 2 and Appendix A). MUFA mainly came from plant oils, accounting for 47.2% (0.8%), followed by pork (24.9% (0.5%)). More than 70.0% of PUFA came from plant foods, particularly plant oils (61.8% (0.2%)).

### 3.2. The Relationship of Dietary Fats and Blood Lipids

A total of 21,539 women (55.1%) were included in the selected sample who were younger (51.1 vs. 52.3 years old, mean), had a little higher dietary energy (1751.0 vs. 1738.8 kcal, mean) but similar dietary fats profile, compared with the whole sample (Appendix A). Within this sample, total fats and fats of various sources and types were inversely associated with energy intake (lowest in the middle quintile), although fats of various sources and types were positively associated with total fat (all *p*_-trend_ < 0.05). Notably, intakes of animal fat and vegetables (energy-adjusted) increased with total fat, SFA, and MUFA, but decreased with PUFA (all *p*_-trend_ < 0.05), while fruit intake, highest in the middle quintile of total fat, animal fat, and SFA, increased with PUFA but decreased with total fat and MUFA (*p* or *p*_-trend_ < 0.05 for above). Moreover, compared with the situation before the median of total fat intake, a higher proportion of MUFA (particularly animal source) and other two types of animal fats but lower proportion of plant PUFA and plant SFA in total fats after the median of total fat intake were detected in further analysis (all *p* < 0.0001, Appendix A). Dietary fiber decreased as various fats increased (all *p*_-trend_ < 0.05). The association of various dietary fats with other characteristics of the selected sample was similar with the whole sample, with the middle quintile of various dietary fats having the lowest physical activity and the least percentage of current smokers and excessive drinkers (*p* or *p*_-trend_ < 0.05 for above).

Generally, total fat had a weaker association with blood lipids than fats of various types and sources (Figure 1 and Appendix A), while animal-source fats, irrespective of the type, had a similar strength and direction in their associations with blood lipids. Plant fats, according to their types, varied in their strength of associations with blood lipids yet with a similar direction. SFA had the strongest association with lipid biomarkers, while MUFA the weakest except with HDL-c and TCHDL. In sensitivity analysis, the exclusion of 4300 participants with diagnosed hypertension did not materially change the results.

Figure 2 shows that plant fats, irrespective of the type, were negatively associated with all blood lipids except HDL-c (positively associated with plant SFA and plant MUFA), contrary to the situation of animal fats (all positively associated, all *p_-_*_trend_ < 0.0001). Total fat was positively associated with TC and LDL-c, but also positively associated with HDL-c, and negatively associated with TGHDL (all *p*_-trend_ < 0.05). While, across the range of as wide as 30%, TE between lowest quintile and highest quintile of total fat, there was only very slight changes in lipid biomarkers (TC, LDL-c, and HDL-c changed only 0.07, 0.05, and 0.03 mmol/L, respectively, Appendix A). SFA was positively associated with TC, LDL-c, HDL-c, and TCHDL, while PUFA were negatively associated with all lipid biomarkers (all *p* < 0.01) other than HDL-c, and MUFA negatively associated with LDL-c and TCHDL, but positively associated with HDL-c (all *p* < 0.01).

Notably, total fat, SFA, and MUFA (even plant MUFA), posited nonlinear relationships with TG and TGHDL which changed directions before and after the median of fat intake (all *p*_-interaction_ < 0.05, Figure 2 and Table 3). Similar were the relationships of MUFA with TC and LDL-c (both *p*_-interaction_ < 0.05), which were different from the nearly linear relationships of PUFA with all lipid biomarkers (*p*_-interaction_ > 0.05, Appendix A). SFA, together with the three types of animal fats, also presented nonlinear relationships with TC, LDL-c, and HDL-c (*p*_-interaction_ < 0.05, Table 3 and Appendix A), but with no changed direction in these relationships.

In the replacement analysis, replacing total fat with carbohydrate resulted in adverse changes in most lipids markers (higher LDL-c, TCHDL, and TGHDL, lower HDL-c, all *p* < 0.05, Table 4) when total fat was low, with the most adverse changes appearing when replacing plant fat or PUFA with carbohydrate, which worsened all lipid biomarkers (all *p* < 0.01), while the most favorable changes appeared when replacing animal fat with plant fat or SFA with PUFA, which, regardless of fat intake level, lowered all lipid biomarkers despite the slightly lowered HDL-c (all *p* < 0.05). Nevertheless, replacement of animal fat with plant fat or SFA with PUFA more effectively improved lipid biomarkers than with carbohydrate: they lowered TC, LDL-c, and TG much more, lowered HDL-c less, and additionally lowered TCHDL and TGHDL (Table 5 vs. Table 4). Overall, replacement of SFA with carbohydrate was associated with lowered LDL-c, TC, and HDL, and unchanged TCHDL and TGHDL (all *p* < 0.0001 except TCHDL and TGHDL). However, when total fat was low, this replacement was associated with elevated TCHDL and marginally elevated TGHDL (*p* < 0.05 for TCHDL, *p* = 0.051 for TGHDL). Similar was the situation for the replacement of animal fat with carbohydrate. Additionally, replacement of MUFA with carbohydrate lowered TG and TGHDL significantly only when total fat was high; likewise, replacement of SFA with MUFA elevated TG significantly only when total fat was high (all *p*_-interaction_ < 0.05).

In comparison with the PURE study, we used the systematic review of clinical trials by the World Health Organization (WHO) as the control (Table 6) [26]. Apparently, the present results were closer to the review than the PURE study (no matter in overall or in the China-specific part). The effect of replacing SFA with carbohydrate in the present study was not so harmful as the PURE and the review, especially when total fat was high, while the effect of replacing SFA with unsaturated fats in the present study and the review were more beneficial than the PURE. Compared with the review, the present results yielded a smaller magnitude in reduction of TC, LDL-c, and HDL-c by replacing SFA with carbohydrate or PUFA, and an opposite effect direction of replacing SFA with MUFA on TG.

## 4. Discussion

With national data from the CNHS (2015–2017), this observational study illustrated the high intake of total fat, the dominance of plant fat and MUFA in total fat and pork-source fat in animal fat, among the Chinese population, and the slightly favorable effect of total fat on blood lipids compared with carbohydrate. However, animal fats and SFA, were possibly more harmful than carbohydrate when total fat was high (>34.9% TE). While, regardless of fat intake level, plant fat and PUFA were most favorable, especially when used to replace animal fat and SFA. Notably, SFA and MUFA (even plant MUFA) posited an opposite relationship direction with TG and TGHDL before and after their median intakes, and MUFA showed adverse effects on TG, compared with carbohydrate and SFA, despite the effect only being significant when total fat was high. The present study showed a less harmful effect of replacing SFA with carbohydrate and more beneficial effect of replacing SFA with unsaturated fats than the PURE study.

To our knowledge, this is the first study using national representative data to illustrate the contribution of dietary fats in various types and food sources to total fats among Chinese adults. The findings can help the development of food-based dietary guidelines for this population and trans-ethnic comparisons with other populations. Additionally, this is the first comprehensive study to determine the relationship of dietary fat intakes with lipid biomarkers in the Chinese population, with a focus on the effect of type, source, and intake level of dietary fats on multiple lipid biomarkers, which can help to observe the net effect of various dietary fats on blood lipids. Furthermore, the comparison of the present findings with the WHO review and the PURE study can help in developing population-targeted strategies and guidelines in blood lipids management.

The findings suggest the dietary fat intake of Chinese adults had reached a comparably high level with America and most European countries, yet with significant disparities in types and food sources: higher portion of PUFA for higher plant oils intake, higher proportion of pork-source fats in animal fat, lower SFA from dairy, meat, and meat products, and slightly lower MUFA intake, which is consistent with a previous study and complements some new findings to that one [8,28,29]. However, the high intake of dietary fats appeared to not be associated with an overall deteriorative effect on blood lipids, albeit the very slight increase in TC and LDL-c. A cross-sectional study in Korean adults suggested that total fats were associated with higher odds of abnormality in LDL-c and TC, but lower odds of TG abnormality [30]. Prospective studies in Western populations had suggested that total fat intake was not associated with CVD risk [31,32,33]. Whether such slight increases in LDL-c, the causal risk biomarker of CVD, can be interpreted as the cause of increased CVD incidence or mortality in the Chinese population requires further studies, which is scarce currently.

The present findings suggested the quantity of dietary fat could affect the association of dietary fats of different types and sources with lipid biomarkers. The replacement of SFA and animal fat with carbohydrate showed both harmful and beneficial effects on blood lipids when total fat was low, but more harmful effects when it was high. In comparison, the PURE study only indicated the neutral effect of SFA and advantage of SFA over carbohydrate on blood lipids. Although the PURE study considered the impact of fat intake level, it seriously underestimated the actual intake of dietary fats in the Chinese population (17.7% TE vs. 34.8% TE in present and 33.2% TE (2011) in previous), particularly unsaturated fats, for no assessment of plant oils [8]. Hence, it can yield a more biased estimation in the replacement effect of SFA with carbohydrate and unsaturated fats, and could not exactly be applicable to the Chinese population, particularly the majority who consumed relatively high energy from fat. Despite this, the present results support part of its conclusion that SFA might have a neutral effect on blood lipids when it was replaced by carbohydrate and when total fat intake was low, since the present study did observe a beneficial effect of SFA on lowering TG and TGHDL, and elevating HDL-c, albeit the possible harmful effect on elevating TC and LDL-c. In fact, moderate SFA intake (about 7.5% TE in present study) could represent a more balanced diet, particularly for a population with traditionally high carbohydrate intake and with high prevalence of high TG and low HDL-c [34].

Notably, high fat intake also appeared to affect the relations of MUFA (even plant MUFA) with blood lipids, and undo some of its beneficial effects (lowering TG and TGHDL, for instance) when total fat was high [35]. The WHO review, whose subjects were predominantly from North America and Europe, indicated a significant reduction of TG and TGHDL when replacing SFA with MUFA, but it did not consider the impact of total fat intake, let alone the impact of disparate food patterns across other districts of the world [26]. On one hand, the increased fat intake level has been suggested to be associated with a higher risk of obesity in the Chinese population, which is directly associated with hypertriglyceridemia [36,37]. On the other hand, higher consumption of dietary fats, although most were plant oils, often was accompanied by higher meat intake (pork as the dominance, whose MUFA is in the largest proportion) in the Chinese population, particularly the southern residents [23,38]. This explained why animal MUFA and animal SFA contributed most to the increased percentage of total fat after the median intake, and why MUFA intake (animal and plant) was higher in southern residents. A previous prospective study in the Chinese population indicated that diets of high animal fat (low carbohydrate) were associated with a higher risk of hypertriglyceridemia (odds ratio (OR) = 1.51) [39]. The high intake of SFA and MUFA, of animal source particularly, when total fat was high, could be exactly the result of such a diet. In addition, regardless of the types, the present study showed the same relationship direction of dietary fats in animal or plant sources with lipid biomarkers. Prospective studies in Western countries also suggested the food source of dietary fats might count more in their relationships with CVD risk [12,13,14,31].

Compared with the WHO review and other clinical trials, the present results indicated a similar beneficial effect of replacing saturated fats with PUFA on lipid biomarkers, but with a smaller magnitude [40,41]. In addition to the mentioned disparities in food patterns, cooking methods of plant oils might also play a role, since many Chinese cuisines were stir-fried, which could impair some beneficial effects of PUFA by oxidation [42].

The present study had some limitations. The effect of MUFA on lipid markers could be complexed by SFA, due to the simultaneous occurrence of MUFA and SFA in animal fats. The nonlinear relationship of fat intake with lipid biomarkers could lead to bias of estimation when using a linear-based regression model, although the stratification analysis had been performed. Fatty acids of some mixed foods, particularly processed food, were unable to be accurately estimated for the incomplete data in the Chinese Food Composition tables (FCT). We complemented some of them by estimation from similar foods, and referring to the Japanese FCT and United States Department of Agriculture (USDA) food composition database. Furthermore, we did not assess the impact of trans fats on blood lipids, which might affect the replacement effect of other fats. Lastly, we did not assess the optimal fat intake level for blood lipids management in the Chinese population, which is worth further study.

## 5. Conclusions

Fat intake in the Chinese population had reached quite a high level, but with significant disparities in types and food sources from Western countries and potential neutral effects on blood lipids. Replacement of SFA (animal fat) with PUFA (plant fat) was associated with the most favorable changes in blood lipids, which was also more effective than the replacement SFA (or animal fat) with carbohydrate, which resulted in neutral effects when total fat was low, but a slightly beneficial effect when total fat was high. The quantity and food sources of dietary fats and background food patterns should be emphasized in blood lipids management of the Chinese population, rather than merely types. The present results were probably more applicable to the Chinese population than the PURE study.

## Figures and Tables

**Figure 1 nutrients-14-05262-f001:**
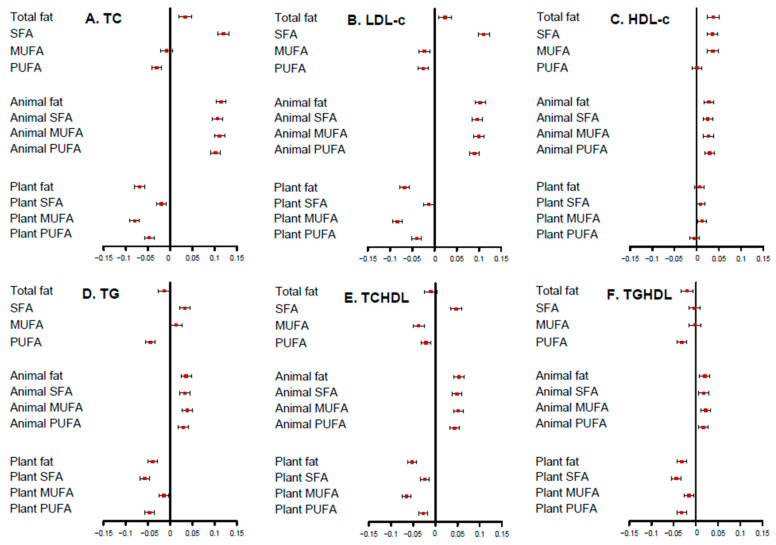
Standardized regression coefficients for the relationships of various fats with lipid biomarkers. Adjusted for total energy, dietary fiber, dietary cholesterol, vegetables, fruits, age, sex, north/south region, urban/rural location, education (categorical), smoking (categorical by status and cigarettes), drinking status (categorical by grams), BMI (continuous), physical activity (continuous), sleep time (continuous), diagnosed hypertension (yes/no), and family history (yes/no). Abbreviation: SFA, saturated fatty acids; MUFA, monounsaturated fatty acids; PUFA, polyunsaturated fatty acids; TC, total cholesterol; LDL-c, low-density lipoprotein; HDL-c, high-density lipoprotein; TG, triglyceride; TCHDL, TC-to HDL-c ratio; TGHDL, TG to HDL-c ratio; BMI, body mass index.

**Figure 2 nutrients-14-05262-f002:**
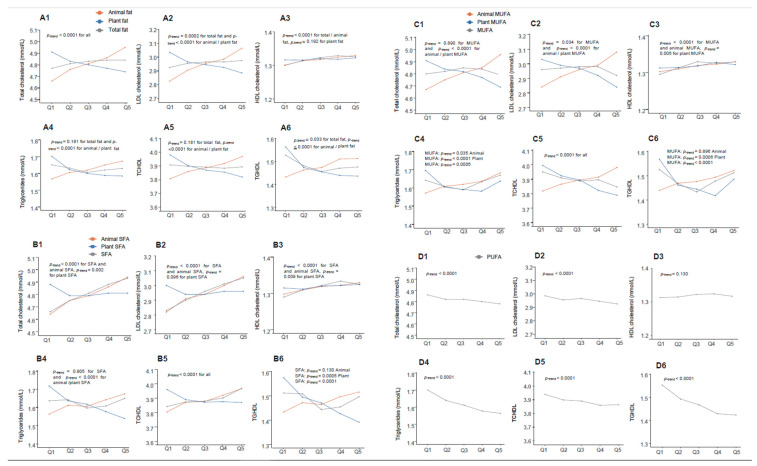
Changing trends of lipid biomarkers across quintiles of various dietary fats. Model adjusted for total energy, dietary fiber, dietary cholesterol, vegetables, fruits, age, sex, north/south region, urban/rural location, education (categorical), smoking (categorical by status and cigarettes), drinking status (categorical by grams), BMI (continuous), physical activity (continuous), sleep time (continuous), diagnosed hypertension (yes/no), and family history (yes/no). (**A1**–**A6**): Total/animal/plant fat; (**B1**–**B6**): SFA and animal/plant SFA; (**C1**–**C6**): MUFA and animal/plant MUFA; (**D1**–**D6**): PUFA.

**Table 1 nutrients-14-05262-t001:** Age-sex-standardized intakes of dietary energy, fat, and fats of various types and sources in Chinese adults with different characteristics (*n* = 48,315) ^1^.

	Number of Subjects (%)	Energy, kcal	Total Fat		Animal Fat		SFA		MUFA		PUFA	
			g	% kcal	g	% kcal	g	% kcal	g	% kcal	g	% kcal
All participants	48,375 (100.0)	1668.9 (1335.4–2066.1) ^2^	63.6 (45.1–86.6)	34.8 (27.5–43.6)	22.1 (10.2–38.3)	12.3 (6.0–19.9)	14.3 (9.8–20.3)	7.9 (5.8–10.2)	23.5 (15.6–34.2)	12.7 (9.3–17.1)	15.5 (10.2–23.1)	8.4 (5.9–11.9)
Age group, year												
30.0–44.9	12,306 (25.4)	1661.9 (1328.9–2066.5) *	63.5 (44.9–86.5)	34.7 (27.8–42.7)	24.3 (12.3–40.8)	13.5 (7.2–21.1) *	14.8 (10.1–20.8)	8.1 (6.0–10.5) *	23.4 (15.6–34.1)	12.8 (9.5–16.9)	14.8 (9.8–22.1)	8.1 (5.8–11.4) †
45.0–59.9	22,394 (46.3)	1698.6 (1357.3–2088.0) *	64.9 (46.2–87.9)	35.0 (27.5–43.7)	21.3 (9.6–37.2)	11.7 (5.5–19.2) *	14.3 (9.8–20.3)	7.7 (5.8–10) *	23.9 (16.0–34.8)	12.7 (9.3–17.3)	16.2 (10.7–24.2)	8.8 (6.0–12.3) †
60.0–69.9	13,675 (28.3)	1624.2 (1301.8–2008.8) *	60.5 (42.9–83.2)	34.7 (26.5–43.3)	17.9 (7.3–33.2)	10.3 (4.4–17.9) *	12.9 (8.7–18.5)	7.3 (5.3–9.7) *	22.4 (14.8–33.2)	12.6 (8.9–17.3)	15.7 (10.3–23.0)	8.7 (6.1–12.2) †
Sex												
Male	21,679 (44.8)	1864.1 (1500.8–2278.3) †	70.2 (50.5–94.6)	34.5 (27.6–43.1) †	26.0 (12.5–43.8)	13.0 (6.5–20.8) †	16.1 (11.1–22.6)	7.9 (5.9–10.3) †	26.2 (17.6–37.8)	12.8 (9.5–17.2)	16.9 (11.1–25.1)	8.2 (5.8–11.6) †
Female	26,696 (55.2)	1517.3 (1225.8–1848.0) †	57.8 (41.1–78.2)	35.1 (27.4–43.3) †	18.9 (8.8–33.2)	11.8 (5.6–19.2) †	12.8 (8.7–18.0)	7.8 (5.7–10.1) †	21.4 (14.1–30.7)	12.7 (9.2–17.1)	14.4 (9.5–21.2)	8.6 (6–12.1) †
Location												
Urban	19,981 (41.3)	1628.7 (1306.2–2023.7) †	62.5 (45.1–85.9)	35.2 (28.5–43)	23.9 (12.4–38.5)	13.5 (7.5–20.3) †	14.8 (10.4–20.4)	8.2 (6.3–10.5) †	23.1 (15.6–33.2)	12.7 (9.6–16.8)	15.2 (10.1–22.4)	8.5 (6–11.6)
Rural	28,394 (58.7)	1716.4 (1366.5–2124.2) †	64.3 (45.1–87.6)	34.3 (26.4–43.4)	19.8 (8.2–38.0)	10.9 (4.6–19.3) †	13.8 (9.1–20.0)	7.4 (5.3–9.8) †	24.0 (15.6–35.6)	12.8 (9–17.5)	15.9 (10.3–23.9)	8.4 (5.7–12.2)
Region												
North	20,090 (41.5)	1606.3 (1272.6–2002.5) †	53.2 (37.4–74.5)	30.8 (23.5–38.8) †	13.7 (5.8–25.7)	8.1 (3.2–14.0) †	11.7 (8.0–16.7)	6.8 (4.9–8.8) †	17.4 (11.5–25.6)	10.0 (7.2–13.2) †	16.2 (10.4–25.0)	9.3 (6.2–13.4) †
South	28,285 (58.5)	1710.8 (1375.3–2100.5) †	69.4 (51.4–92.9)	37.4 (30.1–45.6) †	28.1 (15.3–45.1)	15.1 (8.8–22.9) †	16.1 (11.2–22.3)	8.5 (6.5–10.9) †	27.4 (19.6–39.0)	14.6 (11.2–19.2) †	15.0 (10.1–22.1)	8.1 (5.7–11.1) †
Education												
Illiteracy	6338 (13.1)	1573.2 (1255.3–1953.8) †	58.2 (40.6–80.7)	34.7 (25.6–43.9)	15.0 (5.2–31.5)	8.7 (3.1–17.7) *	11.6 (7.5–17.2)	6.9 (4.7–9.3) *	22.5 (14.4–34.0)	12.9 (8.9–18.7) *	14.8 (9.6–22.3)	8.6 (5.8–12.4) *
Primary school	16,970 (35.1)	1686.4 (1344.4–2085.8) †	64.8 (45.9–88.4)	35.5 (27.6–44.3)	20.4 (8.8–38.2)	11.5 (5.1–20) *	13.8 (9.4–19.9)	7.6 (5.5–10.1) *	24.7 (16.3–37.1)	13.4 (9.6–18.5) *	15.7 (10.3–23.5)	8.5 (5.8–12.1) *
Middle school	15,466 (32.0)	1726.0 (1374.9–2126.6) †	65.5 (46.6–88.2)	34.3 (27.4–42.7)	22.2 (10.4–38.0)	12.1 (6.0–19.1) *	14.7 (10.0–20.5)	7.8 (5.8–9.9) *	23.9 (15.7–34.2)	12.6 (9.2–16.6) *	16.2 (10.8–24.0)	8.5 (6.1–12.1) *
High school	6417 (13.3)	1656.5 (1330.5–2045.0) †	62.0 (44.9–86.4)	34.9 (28–43)	23.5 (13.1–39.0)	13.5 (7.5–20.7) *	15.0 (10.3–21.2)	8.4 (6.3–10.6) *	23.0 (15.4–33.7)	12.6 (9.3–16.6) *	15.3 (10.1–22.9)	8.5 (5.8–11.8) *
College and above	3184 (6.6)	1587.9 (1272.1–1962.8) †	59.7 (43.4–81.9)	34.5 (28.3–41.7)	26.9 (15.8–41.6)	15.6 (10.0–21.7) *	15.3 (11.0–21.0)	8.8 (7.1–10.9) *	21.4 (14.8–29.9)	12.2 (9.5–15.7) *	13.7 (9.1–20.3)	7.9 (5.7–10.8) *
Smoking status												
Never	32,703 (67.6)	1589.6 (1280.6–1954.7) *	60.4 (42.8–82.1)	34.9 (27.2–43.1) *	20.4 (9.4–35.4)	12.0 (5.8–19.4) *	13.5 (9.1–18.9)	7.8 (5.8–10.1) *	22.2 (14.8–32.3)	12.6 (9.2–16.9) *	14.8 (9.8–22.2)	8.6 (5.9–12)
Ever	3057 (6.3)	1833.3 (1476.7–2208.6) *	67.9 (49.1–88.6)	34.0 (27–42.3) *	22.2 (9.7–37.5)	11.4 (5.3–17.9) *	15.4 (10.1–20.9)	7.6 (5.7–9.7) *	25.5 (17.1–35.4)	12.5 (9.2–16.8) *	17.1 (11.5–24.5)	8.3 (6–11.9)
Current, <10 cigarettes/day	2989 (6.2)	1862.8 (1492.2–2230.1) *	69.7 (51.0–93.6)	34.5 (27.9–43.0) *	26.8 (12.3–45.0)	13.9 (6.5–21.5) *	16.0 (11.0–22.9)	8.1 (5.9–10.3) *	26.6 (17.8–37.4)	12.9 (9.6–17.2) *	16.4 (10.9–23.9)	8.0 (5.6–11.3)
Current, 10–19 cigarettes/day	2882 (6.0)	1849.4 (1494.2–2296.6) *	68.9 (49.2–97.2)	34.7 (28.4–43.7) *	26.2 (13.8–43.3)	13.3 (6.6–21.1) *	16.3 (11.0–23.7)	8.1 (6.2–10.9) *	26.3 (17.2–37.8)	12.8 (9.7–17.4) *	16.7 (10.9–25.6)	8.4 (5.8–12)
Current, ≥20 cigarettes/day	6744 (9.4)	1868.2 (1489.6–2309.5) *	72.4 (52.1–97.3)	35.4 (28.2–44.3) *	28.0 (13.3–47.0)	13.9 (6.9–22.0) *	16.4 (11.2–23.1)	8.2 (6.0–10.4) *	27.4 (18.7–39.8)	13.3 (9.9–17.9) *	17.1 (11.2–25.2)	8.2 (5.7–11.6)
Drinking status												
Never	29,758 (61.5)	1593.8 (1277.2–1969.2) *	60.7 (42.9–82.4)	35.0 (27.2–43.6)	20.2 (9.0–36.0)	12.0 (5.5–19.7)	13.3 (9.0–19.1)	7.7 (5.7–10.2) †	22.4 (14.7–32.8)	12.8 (9.2–17.3)	14.9 (9.8–22.1)	8.5 (6–12)
Moderate	12,476 (25.8)	1744.5 (1398.2–2122.2) *	66.5 (47.5–89.3)	34.7 (27.9–42.9)	24.0 (12.3–39.6)	13.0 (6.7–19.9)	15.3 (10.5–21.1)	8.1 (6.1–10.2) †	24.5 (16.6–35.4)	12.7 (9.6–16.9)	15.9 (10.5–24.0)	8.4 (5.8–11.9)
Excessive, 15–24.9 g/day	1599 (3.3)	1791.0 (1475.7–2197.4) *	68.8 (47.5–93.9)	34.1 (25.9–42.4)	25.0 (13.4–43.6)	13.0 (7.1–20.7)	15.7 (10.2–22.4)	7.7 (5.6–9.9) †	25.2 (15.9–36.3)	12.5 (8.6–17.5)	15.5 (10.1–24.8)	7.9 (5.5–11.3)
Excessive, ≥25.0 g/day	4542 (9.4)	1904.7 (1524.7–2352.9) *	71.8 (53.0–94.9)	34.4 (28.1–42.6)	26.9 (11.8–44.8)	12.7 (6.4–20.8)	16.4 (11.7–22.7)	7.9 (6–10.1) †	26.9 (18.2–37.6)	12.7 (9.7–16.8)	17.8 (11.5–25.5)	8.4 (5.8–11.9)
Family history												
Yes	20,404 (42.2)	1669.4 (1338.5–2064.1)	63.0 (45.2–85.5)	34.3 (27.2–42.7)	21.4 (10.6–36.3)	12.1 (6.1–18.9) †	14.2 (9.9–20.1)	7.8 (5.9–10) †	22.8 (15.2–32.7)	12.3 (9–16.4) †	15.9 (10.6–23.8)	8.6 (6.2–12.2) †
No	27,971 (57.8)	1668.2 (1330.9–2067.9)	64.1 (45.0–87.5)	35.3 (27.7–43.7)	22.9 (10.0–39.9)	12.7 (6–20.9) †	14.4 (9.6–20.4)	7.9 (5.8–10.3) †	24.2 (15.9–35.5)	13.2 (9.6–17.7) †	15.1 (9.9–22.6)	8.3 (5.7–11.6) †
Hypertension												
Yes	21,370 (44.2)	1661.7 (1333.3–2062.7)	64.1 (45.7–86.9)	34.1 (26.7–43.2) †	23.6 (11.6–39.6)	11.2 (5–18.5) †	14.7 (10.0–20.7)	7.5 (5.6–9.8) †	23.7 (15.9–34.5)	12.3 (8.9–17)	15.0 (10.0–22.6)	8.7 (6.1–12.3)
No	27,005 (55.8)	1684.0 (1338.2–2073.4)	62.5 (44.4–86.0)	35.3 (27.9–43.2) †	20.0 (8.5–35.8)	13.2 (6.7–20.8) †	13.7 (9.3–19.6)	8.1 (6–10.4) †	23.0 (15.1–33.7)	13.0 (9.6–17.2)	16.1 (10.6–24.0)	8.3 (5.8–11.6)

^1^ Absolute values of total fat and fat of various types and sources between/among various subgroups were not statistically tested. ^2^ Median; interquartile range (IQR) in parentheses (all such values). * *p*_-trend_ < 0.05. Abbreviation: SFA, saturated fatty acids; MUFA, monounsaturated fatty acids; PUFA, polyunsaturated fatty acids. † *p* < 0.05 for between-group comparisons.

**Table 2 nutrients-14-05262-t002:** Age-sex-standardized proportion of various fat sources in various types of fats and total fat (*n* = 48,315) *.

	Food Group	In SFA, %	In MUFA, %	In PUFA, %	In Total Fat, %
Major Animal sources	Pork	30.4 ± 0.5	24.9 ± 0.5	9.8 ± 0.3	18.9 ± 0.5
	Other meats	2.5 ± 0.1	1.5 ± 0.1	0.8 ± 0.0	1.5 ± 0.1
	Processed meat	1.3 ± 0.1	1.1 ± 0.1	0.5 ± 0.0	1.0 ± 0.1
	Poultry	3.4 ± 0.2	3.2 ± 0.3	1.9 ± 0.2	2.6 ± 0.2
	Fish	1.2 ± 0.1	1.1 ± 0.1	1.0 ± 0.1	1.0 ± 0.1
	Dairy and eggs	9.6 ± 0.3	3.2 ± 0.1	1.1 ± 0.0	3.9 ± 0.1
Major Plant sources	Nuts and soybeans	5.7 ± 0.2	4.7 ± 0.2	8.9 ± 0.2	5.6 ± 0.1
	High MUFA oils	6.6 ± 0.5	17.0 ± 1.1	16.3 ± 1.1	13.5 ± 0.9
	MUFA–PUFA balanced oils	11.5 ± 0.7	16.3 ± 1.0	21.7 ± 1.4	14.8 ± 0.9
	High PUFA oils	11.1 ± 0.7	13.9 ± 0.9	23.8 ± 1.3	15.5 ± 0.9
	High SFA oils	0.0 ± 0.0	0.0 ± 0.0	0.0 ± 0.0	0.0 ± 0.0
Other		16.7 ± 0.5	13.0 ± 0.4	14.3 ± 0.4	21.7 ± 0.5

* Values in the table were expressed as mean ± standard error (SE).

**Table 3 nutrients-14-05262-t003:** Beta coefficient (95% CI) for change in lipid biomarkers for per 11 g increment of total fat, saturated fat, and (animal/plant) monounsaturated fat, stratified by median intake *.

Fats	Lipid Markers	<Median, E%	≥Median, E%	*p* _-interaction_
Total fat	TC, mmol/L	0.015 (0.003–0.027)	−0.001 (−0.009–0.007)	0.005
	LDL-c, mmol/L	0.008 (−0.002–0.018)	−0.003 (−0.011–0.005)	0.029
	HDL-c, mmol/L	0.007 (0.003–0.011)	0.000 (−0.002–0.002)	0.000
	TG, mmol/L	−0.013 (−0.027–0.001)	0.002 (−0.008–0.012)	0.018
	TCHDL	−0.009 (−0.021–0.003)	−0.004 (−0.014–0.006)	0.345
	TGHDL	−0.022 (−0.038–−0.006)	−0.001 (−0.0128–0.016)	0.012
SFA	TC, mmol/L	0.207 (0.162–0.252)	0.066 (0.044–0.088)	<0.0001
	LDL-c, mmol/L	0.163 (0.124–0.202)	0.057 (0.039–0.075)	<0.0001
	HDL-c, mmol/L	0.041 (0.025–0.057)	−0.001 (−0.009–0.007)	<0.0001
	TG, mmol/L	−0.032 (−0.085–0.021)	0.033 (0.009–0.057)	0.029
	TCHDL	0.057 (0.004–0.110)	0.046 (0.022–0.070)	0.801
	TGHDL	−0.070 (−0.141–0.001)	0.026 (−0.005–0.057)	0.009
MUFA	TC, mmol/L	0.054 (0.027–0.081)	−0.033 (−0.045–−0.021)	<0.0001
	LDL-c, mmol/L	0.027 (0.004–0.051)	−0.034 (−0.046–−0.022)	<0.0001
	HDL-c, mmol/L	0.031 (0.0212–0.041)	0.002 (−0.002–0.006)	<0.0001
	TG, mmol/L	−0.032 (−0.063–−0.001)	0.018 (0.004–0.032)	0.000
	TCHDL	−0.049 (−0.080–−0.018)	−0.031 (−0.045–−0.017)	0.669
	TGHDL	−0.065 (−0.106–−0.023)	0.013 (−0.007–0.033)	0.000
Plant MUFA	TC, mmol/L	−0.094 (−0.127–−0.061)	−0.044 (−0.056–−0.032)	0.010
	LDL-c, mmol/L	−0.068 (−0.097–−0.039)	−0.045 (−0.055–−0.035)	0.381
	HDL-c, mmol/L	0.006 (−0.006–0.018)	0.001 (−0.003–0.005)	0.854
	TG, mmol/L	−0.107 (−0.148–−0.066)	0.017 (0.003–0.031)	<0.0001
	TCHDL	−0.104 (−0.143–−0.065)	−0.034 (−0.048–−0.020)	0.010
	TGHDL	−0.118 (−0.171–−0.065)	0.017 (−0.001–0.035)	<0.0001

* The median of total fat, SFA, and MUFA were 68.7, 15.4, and 25.7 g, corresponding to 34.9%, 7.6%, and 12.9% of total energy, respectively. Models were adjusted for total energy, dietary fiber (energy adjusted), dietary cholesterol (energy adjusted), vegetables (energy adjusted), fruits (energy adjusted), age, sex, north/south region, urban/rural location, education (categorical), smoking (categorical by status and cigarettes), drinking status (categorical by grams), BMI (continuous), physical activity (continuous), sleep time (continuous), diagnosed hypertension (yes/no), and family history (yes/no). Abbreviation: CI, confidence interval.

**Table 4 nutrients-14-05262-t004:** Changes in lipid biomarkers when dietary fats were replaced by carbohydrate, stratified by fat intake level *.

Replacement	Lipid Biomarkers	Overall	*p*	<Median	>Median	*p* _-interaction_
Total fat to carbohydrate	TC, mmol/L	0.004 (−0.002–0.010)	0.091	0.000 (−0.009–0.008)	0.009 (0.001–0.018)	0.094
	LDL-c, mmol/L	0.009 (0.005–0.013)	0.000	0.010 (0.003–0.018)	0.011 (0.003–0.018)	0.967
	HDL-c, mmol/L	−0.005 (−0.007–−0.003)	<0.0001	−0.011 (−0.013–−0.008)	0.000 (−0.003–0.003)	<0.0001
	TG, mmol/L	0.004 (−0.002–0.010)	0.206	0.007 (−0.003–0.017)	−0.004 (−0.014–0.006)	0.095
	TCHDL	0.017 (0.011–0.023)	<0.0001	0.028 (0.018–0.037)	0.008 (−0.002–0.017)	0.002
	TGHDL	0.008 (0.000–0.016)	0.041	0.017 (0.004–0.029)	−0.002 (−0.015–0.010)	0.028
Animal fat to carbohydrate	TC, mmol/L	−0.017 (−0.023–−0.011)	<0.0001	−0.019 (−0.028–−0.010)	−0.012 (−0.021–−0.003)	0.224
	LDL-c, mmol/L	−0.006 (−0.012–0.000)	0.042	−0.002 (−0.010–0.006)	−0.004 (−0.012–0.004)	0.753
	HDL-c, mmol/L	−0.008 (−0.01–−0.006)	<0.0001	−0.013 (−0.016–−0.010)	−0.003 (−0.006–0.000)	<0.0001
	TG, mmol/L	−0.013 (−0.021–−0.005)	0.001	−0.007 (−0.017–0.003)	−0.020 (−0.031–−0.010)	0.049
	TCHDL	0.007 (0.000–0.014)	0.053	0.019 (0.009–0.030)	−0.002 (−0.012–0.009)	0.001
	TGHDL	−0.005 (−0.015–0.005)	0.266	0.005 (−0.008–0.019)	−0.015 (−0.029–−0.002)	0.017
Plant fat to carbohydrate	TC, mmol/L	0.012 (0.006–0.018)	<0.0001	0.004 (−0.005–0.012)	0.019 (0.010–0.027)	0.009
	LDL-c, mmol/L	0.015 (0.011–0.019)	<0.0001	0.013 (0.006–0.021)	0.018 (0.011–0.025)	0.364
	HDL-c, mmol/L	−0.005 (−0.007–−0.003)	<0.0001	−0.010 (−0.013–−0.008)	0.000 (−0.002–0.003)	0.594
	TG, mmol/L	0.009 (0.003–0.015)	0.003	0.009 (0.000–0.019)	0.002 (−0.008–0.012)	0.055
	TCHDL	0.022 (0.016–0.028)	<0.0001	0.030 (0.020–0.040)	0.013 (0.004–0.023)	0.011
	TGHDL	0.013 (0.005–0.021)	0.002	0.019 (0.006–0.031)	0.003 (−0.010–0.015)	0.064
SFA to carbohydrate	TC, mmol/L	−0.040 (−0.05–−0.03)	<0.0001	−0.037 (−0.047–−0.027)	−0.039 (−0.050–−0.027)	0.794
	LDL-c, mmol/L	−0.018 (−0.026–−0.01)	<0.0001	−0.013 (−0.022–−0.004)	−0.020 (−0.030–−0.01)	0.148
	HDL-c, mmol/L	−0.015 (−0.019–−0.011)	<0.0001	−0.018 (−0.022–−0.015)	−0.010 (−0.014–−0.006)	<0.0001
	TG, mmol/L	−0.007 (−0.017–0.003)	0.217	−0.003 (−0.015–0.009)	−0.014 (−0.027–0.000)	0.116
	TCHDL	0.007 (−0.003–0.017)	0.182	0.017 (0.005–0.029)	−0.007 (−0.020–0.007)	0.000
	TGHDL	0.004 (−0.010–0.018)	0.548	0.013 (−0.002–0.027)	−0.006 (−0.024–0.011)	0.042
MUFA to carbohydrate	TC, mmol/L	0.022 (0.014–0.030)	<0.0001	0.013 (0.004–0.023)	0.030 (0.020–0.040)	0.003
	LDL-c, mmol/L	0.034 (0.028–0.040)	<0.0001	0.030 (0.021–0.038)	0.038 (0.029–0.047)	0.098
	HDL-c, mmol/L	−0.011 (−0.013–−0.009)	<0.0001	−0.015 (−0.018–−0.012)	−0.005 (−0.008–−0.001)	<0.0001
	TG, mmol/L	−0.017 (−0.025–−0.009)	0.000	−0.010 (−0.021–0.002)	−0.025 (−0.037–−0.013)	0.041
	TCHDL	0.044 (0.036–0.052)	<0.0001	0.048 (0.037–0.059)	0.035 (0.023–0.046)	0.047
	TGHDL	−0.007 (−0.019–0.005)	0.232	0.003 (−0.012–0.017)	−0.018 (−0.034–−0.003)	0.015
PUFA to carbohydrate	TC, mmol/L	0.011 (0.003–0.019)	0.006	0.005 (−0.004–0.014)	0.018 (0.007–0.028)	0.028
	LDL-c, mmol/L	0.017 (0.011–0.023)	<0.0001	0.016 (0.008–0.024)	0.020 (0.010–0.029)	0.485
	HDL-c, mmol/L	−0.007 (−0.009–−0.005)	<0.0001	−0.012 (−0.015–−0.008)	−0.002 (−0.005–0.002)	<0.0001
	TG, mmol/L	0.023 (0.013–0.033)	<0.0001	0.022 (0.010–0.033)	0.017 (0.005–0.030)	0.524
	TCHDL	0.024 (0.014–0.034)	<0.0001	0.031 (0.021–0.042)	0.014 (0.002–0.026)	0.008
	TGHDL	0.024 (0.012–0.036)	<0.0001	0.028 (0.013–0.042)	0.014 (−0.002–0.030)	0.119

* Adjusted for total energy, dietary fiber, dietary cholesterol, vegetables, fruits, age, sex, north/south region, urban/rural location, education (categorical), smoking (categorical by status and cigarettes), drinking status (categorical by grams), BMI (continuous), physical activity (continuous), sleep time (continuous), diagnosed hypertension (yes/no), and family history (yes/no).

**Table 5 nutrients-14-05262-t005:** Changes in lipid biomarkers when animal fats or SFA were replaced by plant fat or MUFA/PUFA, stratified by fat intake level *.

Replacement	Lipid Biomarkers	Overall	*p*	<Median	>Median	*p* _-Interaction_
Animal fat to plant fat	TC, mmol/L	−0.048 (−0.054–−0.042)	<0.0001	−0.068 (−0.078–−0.058)	−0.045 (−0.051–−0.039)	0.001
	LDL-c, mmol/L	−0.039 (−0.043–−0.035)	<0.0001	−0.052 (−0.062–−0.042)	−0.037 (−0.043–−0.031)	0.004
	HDL-c, mmol/L	−0.002 (−0.004–0.000)	0.007	−0.006 (−0.01–−0.002)	−0.002 (−0.004–0.000)	0.194
	TG, mmol/L	−0.022 (−0.028–−0.016)	<0.0001	−0.035 (−0.047–−0.023)	−0.018 (−0.024–−0.012)	0.003
	TCHDL	−0.032 (−0.038–−0.026)	<0.0001	−0.043 (−0.055–−0.031)	−0.029 (−0.035–−0.023)	0.038
	TGHDL	−0.019 (−0.027–−0.011)	<0.0001	−0.032 (−0.048–−0.016)	−0.015 (−0.023–−0.007)	0.016
SFA to MUFA	TC, mmol/L	−0.160 (−0.178–−0.142)	<0.0001	−0.170 (−0.221–−0.119)	−0.155 (−0.175–−0.135)	0.599
	LDL-c, mmol/L	−0.145 (−0.159–−0.131)	<0.0001	−0.158 (−0.203–−0.113)	−0.141 (−0.159–−0.123)	0.228
	HDL-c, mmol/L	−0.001 (−0.007–0.005)	0.658	0.006 (−0.003–0.014)	−0.002 (−0.008–0.004)	0.043
	TG, mmol/L	0.025 (0.005–0.045)	0.012	0.017 (−0.012–0.047)	0.027 (0.007–0.047)	0.048
	TCHDL	−0.122 (−0.142–−0.102)	<0.0001	−0.149 (−0.178–−0.120)	−0.118 (−0.138–−0.098)	0.017
	TGHDL	0.021 (−0.004–0.046)	0.109	−0.007 (−0.034–0.042)	0.024 (−0.002–0.050)	0.240
SFA to PUFA	TC, mmol/L	−0.140 (−0.154–−0.126)	<0.0001	−0.157 (−0.182–−0.136)	−0.131 (−0.150–−0.120)	0.036
	LDL-c, mmol/L	−0.113 (−0.125–−0.101)	<0.0001	−0.123 (−0.147–−0.099)	−0.106 (−0.122–−0.09)	0.106
	HDL-c, mmol/L	−0.009 (−0.013–−0.005)	0.000	−0.011 (−0.019–−0.003)	−0.008 (−0.013–−0.003)	0.523
	TG, mmol/L	−0.047 (−0.065–−0.029)	<0.0001	−0.067 (−0.094–−0.04)	−0.042 (−0.060–−0.024)	0.076
	TCHDL	−0.083 (−0.099–−0.067)	<0.0001	−0.090 (−0.121–−0.059)	−0.088 (−0.110–−0.066)	0.344
	TGHDL	−0.034 (−0.056–−0.012)	0.002	−0.013 (−0.054–0.028)	−0.054 (−0.081–−0.027)	0.263

* Adjusted for total energy, dietary fiber, dietary cholesterol, vegetables, fruits, age, sex, north/south region, urban/rural location, education (categorical), smoking (categorical by status and cigarettes), drinking status (categorical by grams), BMI (continuous), physical activity (continuous), sleep time (continuous), diagnosed hypertension (yes/no), and family history (yes/no).

**Table 6 nutrients-14-05262-t006:** Comparison with WHO review and PURE study in changes of lipids biomarkers when replacing SFA with other nutrients *.

Replacement		Present, 11 g, Overall	Present, 11 g, <Median	Present, 11 g, >Median	WHO, 4% TE	PURE, Overall, 5% TE	PURE, China, 5% TE
SFA to carbohydrate	TC	−0.040 (−0.050–−0.030)	−0.037 (−0.047–−0.027)	−0.039 (−0.050–−0.027)	−0.164 (−0.188–−0.140)	−0.085 (−0.089–−0.081)	−0.034 (−0.062–−0.007)
	LDL-c	−0.018 (−0.026–−0.010)	−0.013 (−0.022–−0.004)	−0.020 (−0.030–−0.010)	−0.132 (−0.156–−0.108)	−0.093 (−0.097–−0.089)	−0.025 (−0.049–−0.001)
	HDL-c	−0.015 (−0.019–−0.011)	−0.018 (−0.022–−0.015) †	−0.010 (−0.014–−0.006) †	−0.040 (−0.048–−0.032)	−0.029 (−0.031–−0.028)	−0.020 (−0.030–−0.010)
	TG	NS	NS	−0.014 (−0.027–0.000)	0.044 (0.028–0.056)	0.022 (0.017–0.026)	0.053 (0.020–0.085)
	TCHDL	NS	0.017 (0.005–0.029) †	NS	NS	0.120 (0.066–0.174)	0.037 (0.004–0.070)
	TGHDL	NS	0.013 (−0.002–0.027) †	NS	0.056 (0.040–0.072)	0.047 (0.041–0.053)	0.067 (0.026–0.109)
SFA to MUFA	TC	−0.160 (−0.178–−0.142)	−0.170 (−0.221–−0.119)	−0.155 (−0.175–−0.135)	−0.184 (−0.204–−0.160)	−0.133 (−0.143–−0.124)	NS
	LDL-c	−0.145 (−0.159–−0.131)	−0.158 (−0.203–−0.113)	−0.141 (−0.159–−0.123)	−0.168 (−0.188–−0.148)	−0.182 (−0.192–−0.173)	−0.010 (−0.056–−0.036)
	HDL-c	NS	NS	NS	−0.008 (−0.016–0.000)	−0.028 (−0.031–−0.024)	NS
	TG	0.025 (0.005–0.045)	NS	0.027 (0.007–0.047) †	−0.016 (−0.028–−0.004)	0.032 (0.021–0.043)	0.094 (0.031–0.158)
	TCHDL	−0.122 (−0.142–−0.102)	−0.149 (−0.178–−0.120) †	−0.118 (−0.138–−0.098) †	−0.108 (−0.132–−0.088)	−0.045 (−0.058–−0.032)	NS
	TGHDL	NS	NS	NS	NS	0.050 (0.036–0.064)	0.085 (0.005–0.164)
SFA to PUFA	TC	−0.140 (−0.154–−0.126)	−0.157 (−0.182–−0.136) †	−0.131 (−0.150–−0.120) †	−0.256 (−0.280–−0.232)	−0.045 (−0.053–−0.036)	−0.048 (−0.075–−0.027)
	LDL-c	−0.113 (−0.125–−0.101)	−0.123 (−0.147–−0.099)	−0.106 (−0.122–−0.09)	−0.220 (−0.244–−0.200)	−0.016 (−0.024–−0.008)	−0.040 (−0.063–−0.016)
	HDL-c	−0.009 (−0.013–−0.005)	−0.011 (−0.019–−0.003)	−0.008 (−0.013–−0.003)	−0.020 (−0.024–−0.012)	−0.026 (−0.029–−0.023)	−0.025 (−0.034–−0.015)
	TG	−0.047 (−0.065–−0.029)	−0.067 (−0.094–−0.04)	−0.042 (−0.060–−0.024)	−0.040 (−0.056–−0.028)	0.016 (0.006–0.025)	0.064 (0.032–0.096)
	TCHDL	−0.083 (−0.099–−0.067)	−0.090 (−0.121–−0.059)	−0.088 (−0.110–−0.066)	−0.136 (−0.160–−0.112)	0.035 (0.025–0.047)	0.034 (0.002–0.066)
	TGHDL	−0.034 (−0.056–−0.012)	−0.013 (−0.054–0.028)	−0.054 (−0.081–−0.027)	−0.020 (−0.036–−0.008)	0.034 (0.022–0.046)	0.082 (0.042–0.122)

* 11 g SFA were approximately 4% TE of WHO review subjects, who were mostly from North America and Europe; The PURE study did not report the mean energy intake of its participants, rather, we estimated the equivalent percentage energy of 11 g SFA from another PURE study: about 5% TE in 2000 kcal [27]; the color of orange/blue indicated β coefficients in positive/negative values, respectively, the darker the color, the absolute value of β coefficients the bigger. † *p*_-interaction_ < 0.05 in stratification analyses. Abbreviation: TE, total energy; WHO, World Health Organization; PURE, prospective urban rural epidemiology; NS, not significant.

## Data Availability

The data are not allowed to be disclosed according to the National Institute for Nutrition and Health, Chinese Center for Disease Control and Prevention.

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
