# Peer review of "Dietary Fat Intake among Chinese Adults and Their Relationships with Blood Lipids: Findings from China Nutrition and Health Surveillance and Comparison with the PURE Study"

_nutrients, 2022, doi:10.3390/nu14245262_

Round 1
Reviewer 1 Report
1. Figure 1 and 1s are not clear.
2. Please check P values in all tables. It seems some of P values in the main and supplementary files have errors. I highlighted some of them in the supplementary and send as attached. Based on the revelant CI they supposed to be significant or no significantly opposite to your report. Please check all tables of main file in this regard.
3. Add Sd to the mean level in the table 6s.

Author Response
Dear reviewer,
Thanks for your careful inspection and valuable suggestions. Here I sended my responses as attached.
Sincerely,
Rongping Zhao

Reviewer 2 Report
This study is well designed. However, the following points should be improved before publication.
L4: What is “PURE study”? Full name of “PURE” should be incorporated.
L17: “inaccurate assessment” What do you mean?
The following issues in this study should be considered in the discussion.
L27: Replacing total fat with “carbohydrate”:
How about starch? or sugar (sucrose)?
L105-107: How about cholesterol contents of fats?
Author Response
Dear reviewer,
Thanks for your careful inspection and suggestions. Here I send my responses as attached.
Sincerely,
Rongping Zhao

Reviewer 3 Report
In this comprehensive study, the authors assessed patterns of dietary fat intake in a large cohort of adult Chinese population, explored the association of various dietary fats with blood lipid biomarkers and compared the results with other relevant research.
The study is well designed, the results are clearly presented and the discussion is supported by the results. The study is of importance for the field.
Specific comments:
Statistical analysis: this section should be complemented by more detailed description of residual energy adjustment method (results presented in Table 3).
Selection of adequate clinical trials for comparison should be presented in details.
Author Response
Dear reviewer,
Thanks for your suggestions and careful inspection. Here I provided my responses as attached.
Sincerely,
Rongping Zhao

Reviewer 4 Report
The study presented to me for review by Rongping Zhao et al. was designed to assess the profile of fat intake in the Chinese population and comprehensively examine the associations of fats with lipid biomarkers across different types and sources of fats.
The introduction although brief shows a research gap, which the authors decided to exploit in their study.
The material and methods chapter is described correctly. A flowchart of the study's sample selection would have been helpful to improve the perception of perception.
In the study, the authors clearly and logically formulated the objectives of the study, consistently carried out the tasks set, and presented balanced conclusions. There is nothing left to do but congratulate the authors on a well-invited and well-designed survey.
From the technical notes, please only change the way of citation according to ACS style and put references in square brackets.
Author Response
Dear reviewer,
Thanks for your generous appreciation and encouragement. We already changed the reference format, in line with the MDPI format guidelines.
Sincerely,
Rongping Zhao